# New Score for Predicting Results after Catheter Ablation for Atrial Fibrillation: VAT-DHF

**DOI:** 10.3390/jcm13010061

**Published:** 2023-12-22

**Authors:** Alexandrina Nastasă, Ștefan Bogdan, Corneliu Iorgulescu, Andrei Dan Radu, Luminița Craițoiu-Nirlu, Radu Gabriel Vătășescu

**Affiliations:** 1Cardiology Departament, Elias Universitary Emergency Hospital, 011461 Bucharest, Romania; alexandrina.nastasa@yahoo.com (A.N.); stefan_n_bogdan@yahoo.com (Ș.B.); nirluluminita@gmail.com (L.C.-N.); 2Faculty of Medicine, Carol Davila University of Medicine and Pharmacy, 050474 Bucharest, Romania; radu_dan_andrei@yahoo.com; 3Clinial Emergency Hospital Bucharest, 014461 Bucharest, Romania; iorgulescu_corneliu@yahoo.com

**Keywords:** atrial fibrillation, catheter ablation, ablation outcomes, AF recurrences, risk scores

## Abstract

Introduction: Catheter ablation (CA) for atrial fibrillation (AF) has been proven to have the highest efficacy in maintaining sinus rhythm. Several studies have proposed different scores for predicting post-procedural success, but most have not been validated in prospective cohorts. Further research is required to determine the optimal formulae. Purpose: This study aimed to identify independent predictors of AF recurrence after CA and develop a composite score. Methods: Consecutive patients with persistent and paroxysmal AF who underwent CA were retrospectively analyzed. The independent predictors of recurrence were used to create a new predictive score. Results: The cohort included 263 patients with a follow-up of 37.6 ± 23.4 months. Persistent AF, f-waves < 0.1 mV, indexed left atrium volume, the presence of type 2 diabetes, and smaller height were independent predictors of recurrence and were used to create a new scoring model, VAT-DHF (V = Volume, AT = AF Type, D = Diabetes, H = Height, F = f waves). The ROC curve for this new score showed an AUC of 0.869, *p* < 0.0001, 95% CI [0.802–0.936], while those for APPLE and CHA_2_DS_2_-VASc showed an AUC of 0.765, 95% CI [0.637–0.893] and an AUC of 0.655, 95% CI [0.580–0.730], respectively. Patients who had a VAT-DHF score between 0 and 3.25, 3.25 and 6, and ≥6, had success rates of 95.7%, 76.3%, and 25% (*p* < 0.0001), respectively. Conclusions: The novel VAT-DHF score is easy to calculate and may be a useful clinical tool for identifying patients with a low, intermediate, or high risk of AF recurrence after CA.

## 1. Introduction

Catheter ablation (CA) for atrial fibrillation (AF) has been proven to have the highest efficacy in maintaining sinus rhythm [1,2]; however, success rates vary widely, ranging between 29 and 90% [2,3,4]. Not infrequently, cases of very early or late recurrences still occur. Retrospective analyses identified several independent predictors, such as AF type and duration [5], size of left atrium [5], older age [6], female sex [7], number of antiarrhythmics administered before ablation [8], higher values of CHADS_2_ and CHA_2_DS_2_-VASc scores [9], hypertension [2], obstructive sleep apnea syndrome [10], coronary atherosclerosis [4], or metabolic syndrome [11] to be associated with intervention success. Procedural costs are not negligible, and a low but present rate of severe complications exists. Therefore, an accurate estimation of the results is necessary to assess the risk–benefit ratio. The practice of using prediction scores for management of atrial fibrillation is well known since the thromboembolic risk scores and it is still a relevant tool nowadays, with a trend of developing more specific scores, either for subtypes of AF (as with the FLAME [12] score) or for specific therapeutic targets (as when predicting LVEF improvement after ablation [13]). In one of the most recent debates in the field, during the EHRA Congress 2023, Dr. Marco Bergonti of the Cardiocentro Ticino Institute in Lugano, Switzerland, said “further evidence is needed to help stratify and identify those patients who will most likely benefit from atrial fibrillation ablation” as they presented a new score for predicting AF ablation outcomes in a subgroup of patients with heart failure, the ANTWOORD Study [13]. Utilization of this type of scoring system is helpful in taking a more personalized approach in the decision-making process.

Several other studies have proposed different composite scores for predicting post-procedural success, such as APPLE [14], CAAP-AF [15], MB-LATER [16], and SUCCESS [17], but most have not been validated in prospective cohorts [18]. Moreover, some proposed scores included only intra-procedural or post-ablation variables, with poor predictive power and/or low sensitivity/specificity ratios. Therefore, further research is warranted to determine the optimal formula [18].

The primary objective of our study was to identify independent predictors of AF recurrence after CA, to elaborate an easily calculable score, and to compare it with previously described scores.

## 2. Materials and Methods

We retrospectively analyzed 263 consecutive patients with paroxysmal or persistent AF who underwent catheter ablation at our center. Preprocedural characteristics (significant comorbidities, cardiovascular risk factors, AF history, and prior medications), ECGs, routine laboratory tests, and echocardiographic data were collected. ECG measurements included the duration and amplitude of the p-wave, suggestive elements of left atrium overload, a notch in DII, or a negative component >40 ms in V1. The “f” fibrillatory waves were divided into 2 categories, lesser and greater than 0.1 mV.

Patients with paroxysmal AF underwent radiofrequency (RF) pulmonary antral vein isolation (PVAI). Entrance and exit pulmonary vein blocks were demonstrated by pacing maneuvers in sinus rhythm (SR). In patients with persistent AF, a stepwise strategy was used. After the initial PVAI, if SR was not restored, additional lines, or activation-guided ablation of the residual atrial tachycardia/flutter was performed until SR was restored. If this was not possible, a chemical or electrical conversion to SR was performed. A cavotricuspid isthmus (CTI) line was created in all patients at the endpoint of a bidirectional conduction block. During the repeat procedure, the persistence of pulmonary vein isolation was evaluated. In the presence of conduction recovery, re-isolation of the PVs was performed using a strategy similar to that of the initial procedure.

During follow-up, screening for arrhythmia recurrence was performed via clinical interview and 48 h Holter monitoring at 1, 3, and 6 months and every 6 months thereafter. A 3-month blanking period was used to define recurrence status. Recurrence was defined as documented AF/atrial tachycardia/atrial flutter on ECG or 48 h Holter monitoring lasting >30 s.

SPSS Statistics version 22 (IBM) and Analyze-It software for Microsoft Excel were used to analyze the data. Continuous variables were presented as mean ± standard deviation or median (IQR), and categorical variables were presented as frequencies. For comparison of subgroups with and without arrhythmia relapse, the Wilcoxon–Mann–Whitney U test, *t*-test, chi-squared, and Fisher’s exact tests were employed as necessary. Variables that were significantly associated with recurrence risk according to the univariate analysis were introduced into a proportional hazard risk ratio Cox analysis. The cut-off of the *p* value used for inclusion in the multivariable regression model was *p* ≤ 0.05. Cox proportional regression using the Enter Forced Method of the tested variables (including multicollinearity testing (tolerance less than 0.1 and VIF value greater than 10)) was used to validate predictors for time to atrial fibrillation recurrence. Performance of the predicted model was assessed by ROC curve and the ROC curve of the model had superior prediction power compared to the ones for each predictor separately. Variables that were statistically significant according to multivariable Cox analysis were used to create a new prediction score after empirically assigning different weights based on the coefficients obtained in the regression and the degree of separation between the Kaplan–Meier curves. Receiver operating characteristic (ROC) curves were generated for a graphical illustration of the performance of the VAT-DHF, CHA_2_DS_2_-VASc, and APPLE scores in predicting rhythm outcome, with the area under the curve (AUC) equivalent to the *c* index for determining the predictive value for a score. The *c* indices (i.e., areas under the ROC curves) were used to test the new scoring model for predicting post-procedural outcomes.

## 3. Results

The cohort included 263 patients: 65% male, with a mean age of 55.5 ± 11.7 years, and 61% with paroxysmal AF (161 patients).

The mean follow-up period was 37.6 ± 23.4 months, and the mean procedure number was 1.5 ± 0.8. The mean time from the first AF diagnosis to the first procedure was 4.56 ± 3.79 years, with a minimum of 4 months and a maximum of 22 years. Echocardiographic findings showed that most patients had enlarged atria (left rather than right), with a mean indexed left atrium volume of 37.9 mL (moderate dilation category). The ejection fraction of the left ventricle was most often preserved, with a minimum value of 15%, a maximum of 74%, and a mean of 55%.

The overall success rate at the end of the follow-up period (approximately 3 years) after all procedures was 67%. The univariate analysis of the predictors of recurrence is presented in Table 1.

Subsequently, the variables that reached statistical significance in the univariate analysis were introduced into the Cox multivariable regression model. The cut-off of the *p* value used for inclusion in the multivariable regression model was *p* ≤ 0.05. Cox proportional regression using the Enter Forced Method of the tested variables indicated that five of them were independent predictors of recurrence, as shown in Table 2: persistent subtype of AF, fibrillatory f-waves < 0.1 mV, indexed left atrial volume, presence of type 2 diabetes, and smaller height. Diagnosis of the model: CI for exp (B) 95%, probability for variable entry in the model: *p* value less than 0.05, model chi square 52,783, model −2 Log Likelihood 362,369.

ROC analysis showed that patients with a height <170 cm were more likely to experience arrhythmia recurrence (sensitivity of 75% and specificity of 68.2%, AUC = 0.62; 95% CI: 0.51–0.72, *p* = 0.026). This was also confirmed by Fisher’s exact test that showed that height <170 cm is associated with an increased risk of recurrence (*p* = 0.04). Pearson’s chi-squared test for different left atrium volume categories (Table 3) was also statistically significant (*p* < 0.000). Based on the coefficients obtained in the Cox regression (Table 2) and the degree of separation of the Kaplan–Meier curves for each predictor over time (graphic representations are available in the Appendix A), an empirically derived point-based weighting scheme was developed. The variables with the largest separation on the Kaplan–Meier curves over time were assigned the highest number of points in the scoring system. For LAVI, we considered a gradually increasing scheme with one point for each severity class, according to the guideline’s classification of mild/moderate and severe dilatation. The process was facilitated by generating the ROC curves for each of the predictors separately, thus illustrating the increase in prediction power brought by the model score (compared with each predictor individually) as seen in Figure 1 and Table 3.

The VAT-DHF scores were calculated as shown in Table 4. The indexed atrial volume was divided into four categories (according to the echocardiographic classifications of normal, mild, moderate, and severe dilatations).

A cutoff value of 5875 gives the score a sensitivity of 87% and a specificity of 75%. These values, and the AUC of 0.869 (*p* < 0.0001, 95% CI [0.802–0.936]), were better than those reported in the literature for similar scores as seen in Figure 2, for the APPLE score AUC 0.765, *p* = 0.001, 95% CI [0.637–0.893] and for the CHA2DS2Vasc AUC 0.655, *p* = 0.000, 95% CI [0.580–0.730]. Kaplan–Meier analysis showing arrhythmia-free survival for the two groups of patients with a score above and under the cut-off value is shown in Figure 3.

Figure 4 shows that the patients who registered a VAT-DHF score between 0 and 3.25, 3.25 and 6, and ≥6 had a 95.7, 76.3, and 25% success rate, respectively, after a mean follow-up period of 37 months.

Testing the new score on a randomly selected sample (50% of the cohort) showed that the predictive value was maintained at an AUC of 0.872 (*p* < 0.001, 95% CI 0.766–0.942).

## 4. Discussion

The univariate analysis in our study confirms most of the factors associated with AF recurrence postulated in previous studies, such as the type of persistent fibrillation, hypertension, size of the left atrium, and ischemic heart disease (more precisely, those who underwent a myocardial revascularization procedure). It also showed that some established predictors are not confirmed, such as age and female gender (possibly because the group included a smaller number of women) or are at the limit of significance, for example, obstructive sleep apnea. The presence of valvular heart disease (moderate or severe mitral and tricuspid regurgitation, and moderate or severe aortic stenosis) was strongly associated with the risk of recurrence, as well as other structural heart diseases. Among the electrocardiographic variables, the duration and amplitude of the p-wave were not significantly different between the groups with and without recurrence, but the existence of elements suggestive of left atrial overload (a notch in DII or a negative component >40 ms in V1) exceeded the threshold of statistical significance, and the duration of the QRS complex. The mean QRS duration in the relapsed group was 102.06 ms, while in the non-relapsed group, it was 93.04 ms (*p* = 0.006), which also correlated with the presence of LBBB, which was higher among those who relapsed (16 vs. 1.4%, *p* = 0.002).

The main contribution of this study is that it is based on a few clinical and paraclinical variables that are easily obtained and quantified in daily practice, such as AF type, left atrium volume, size of fibrillation f-waves on the EKG, presence of diabetes, and patient height; it is possible to calculate a score for predicting long-term success after AF CA. All individual clinical variables that were identified as independent predictors of relapse in the present study were also found to be associated with post-ablation outcomes in multiple previous studies, but this particular model of combining them offers a very good predictive power, as shown by the ROC analysis. The comparison of predictive power for each of the variables separately and against the new scoring system we developed shows the clear superiority of the latter (Figure 2). Perhaps one of the less encountered variables associated with recurrence is patient height, for which we found only one previous report that concluded that lower height predisposes to lower success rates [19]. Although there are few previous reports associating patient height with AF CA outcomes, it is plausible that it plays a significant role given the known association between greater height and a higher incidence of AF [20]. Therefore, it can be hypothesized that this clinical entity (AF in taller patients) represents a distinct phenotype of AF less connected to risk factors/comorbidities [21], with a potentially better response to ablative therapy.

Fibrillatory “f” wave size is an easy-to-measure parameter with a sensitivity and specificity of 75% and 73%, respectively, in predicting AF recurrence after ablation [22]. Some authors [23] demonstrate that only 12% of those with f > 0.5 mV in V1 and aVF have post-ablation arrhythmic relapses compared to the opposite group.

Elevated LA dimensions are the most commonly reported negative prognostic parameter [5]. However, a cutoff value that tilts the risk–benefit balance against ablation has not been clearly established. Some authors have proposed an LA diameter >43 mm for inclusion in the APPLE prediction score, a score that was also tested in our study population, but the predictive power was lower than that of the newly formed score, VAT-DHF. A recent meta-analysis [24] of 33 studies proposing 13 prediction models showed that all evaluated scores had a degree of bias, and the favorable results obtained initially, although very promising, were not reproducible later; therefore, the present results also require confirmation in a larger external cohort.

Other studies have shown that obstructive sleep apnea syndrome is associated with worse outcomes after ablation [10] and is an independent predictor of recurrence, a fact that was not confirmed by our group, possibly because the general rule in our preablation protocol was that all patients should be screened for sleep apnea syndrome and treated effectively before the procedure.

Several studies [9,25] have shown that CHADS_2_ and CHA_2_DS_2_-VASc scores correlate with post-ablation results. However, we did not consider it appropriate to include them in the prediction model because they did not provide independent information. These scores were derived from other parameters, such as age, sex, LV dysfunction, hypertension, and factors that were included individually in the multivariate analysis. Although it was designed to predict thromboembolic risk, given that many of its components have been shown to predict post-ablation AF recurrence, the CHA_2_DS_2_-VASc score has been used in several studies with modest predictive value. It was also tested in our study population, and similar results were obtained (AUC of 0.65) compared to the AUC of 0.627 described in the literature, which suggests that our population, although not very large, is representative; thereby encouraging testing of the VAT-DHF score on a larger scale.

In this context, our opinion is that this score could help with the appropriate selection of the patients who will benefit the most and be a good tool for prioritization of a limited resource. Also, our experience is that the more information about possible outcomes patients have (especially for non-urgent procedures), the more content they are after the procedure and the more compliant they are to further evaluation and treatment. For example, if a patient with a score higher than six points is informed he has a 75% chance of arrhythmia recurrence but that the arrhythmic burden could still be decreased or even the mortality reduced in some cases, he will perhaps still choose the procedure and not be very disappointed if arrhythmic recurrence occurs.

## 5. Limitations

Our study had some limitations. The number of patients was limited, and the analysis was conducted retrospectively. Intermittent rhythm monitoring after ablation using only 48 h Holter recording may have underestimated the AF recurrence rate, especially in asymptomatic and undetected AF recurrence. As a future direction, we are planning to verify the new score in a prospective analysis with more cases and external validation. In order to do that, we are now including patients both from our laboratory and from another center in Germany, where one of our fellows is currently in training.

## 6. Conclusions

In conclusion, the novel VAT-DHF score is easy to calculate and may provide a useful clinical tool for identifying patients with low, intermediate, or high risk of AF recurrence. However, further studies are needed to validate the results.

## Figures and Tables

**Figure 1 jcm-13-00061-f001:**
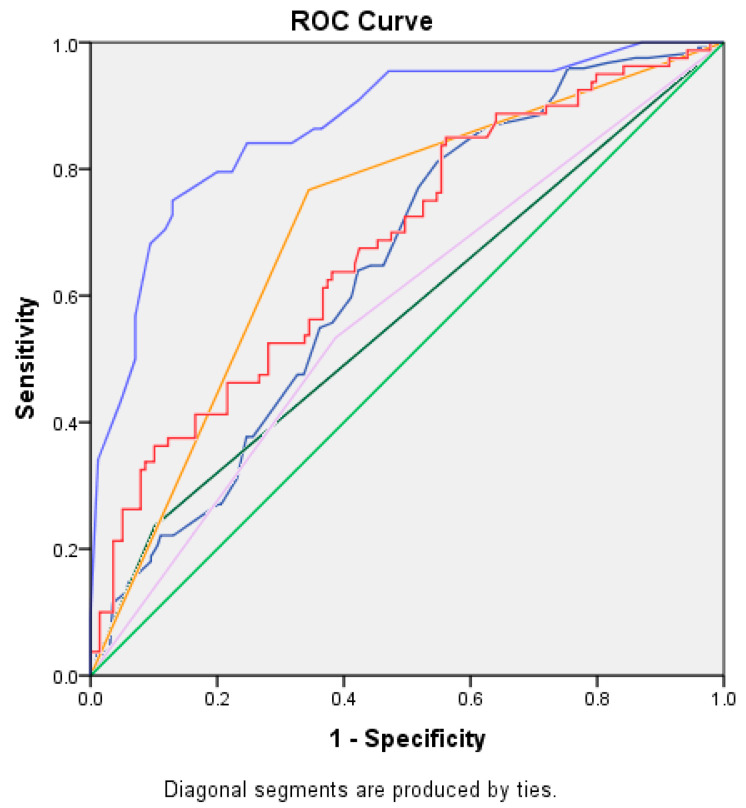
ROC curves for each of the predictors separately. pink: AF type: persistent; dark green: type 2 diabetes; dark blue: height; red: LA indexed volume; orange: f waves < 0.1 mV; blue: model score VAT-DHF.

**Figure 2 jcm-13-00061-f002:**
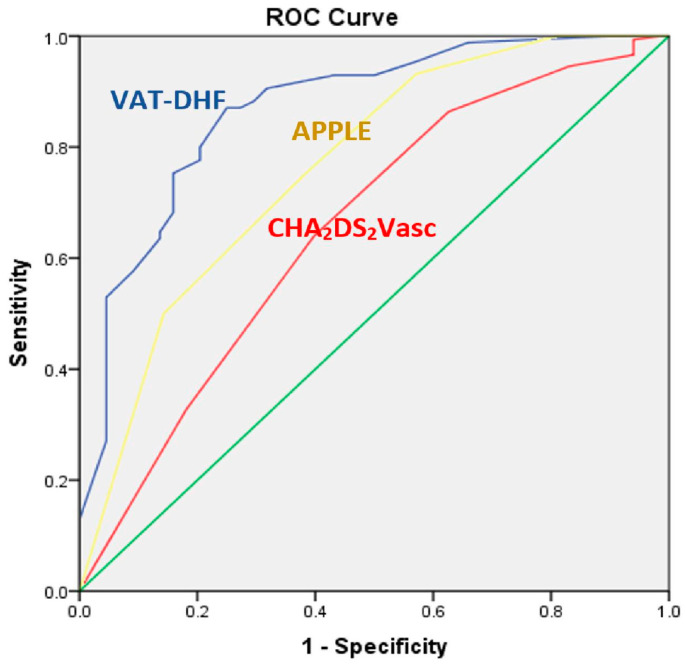
Comparison between ROC curves for VAT-DHF and two other scores used to predict results after atrial fibrillation ablation: CHA_2_DS_2_Vasc and APPLE.

**Figure 3 jcm-13-00061-f003:**
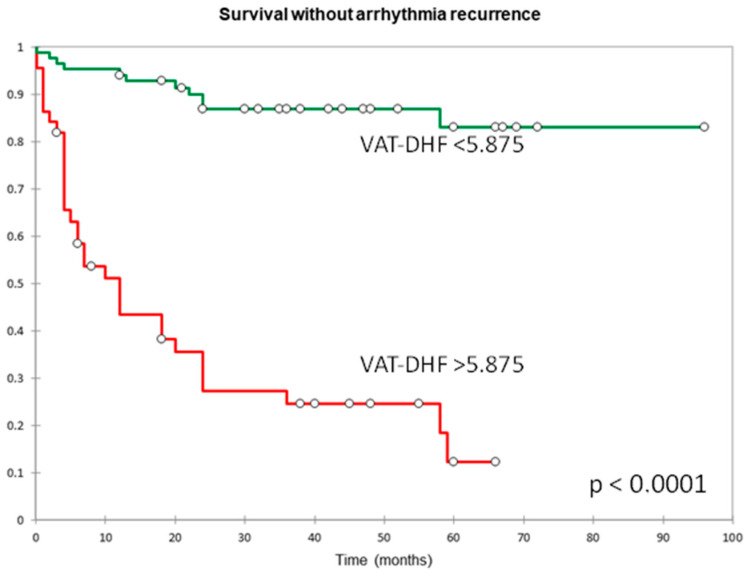
Kaplan–Meier curve showing arrhythmia-free survival for the patients with a score above (green) and under (red) cut-off value of the VAT-DHF score.

**Figure 4 jcm-13-00061-f004:**
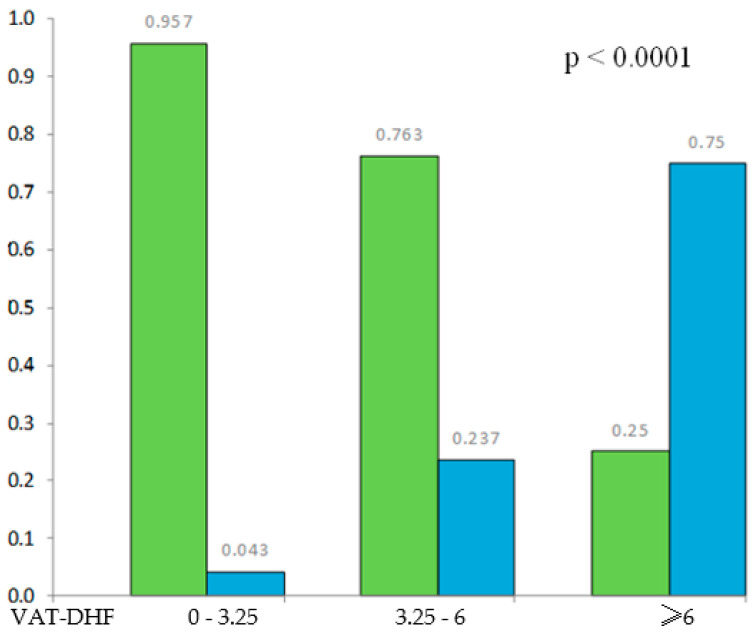
Distribution of AF recurrences within study population according to the VAT-DHF score; Green = no recurrence group; Blue = recurrence group.

**Table 1 jcm-13-00061-t001:** Univariate analysis for variables associated with arrhythmic recurrence.

	without Recurrence	with Recurrence	*p*-Value
**Demographics/history:**			
Male sex	68%	62%	0.37
Age (years)	54.98 ± 11.2	56.56 ± 12.7	0.31
Height (cm)	175.38 ± 10.52	172.16 ± 9.75	0.02
BMI (kg/m^2^)	28.42 ± 3.69	28.79 ± 4.8	0.51
Familial history of AF	5.2%	9.3%	0.2
Arterial hypertension	50%	65.1%	0.02
Ischemic heart disease	10.8%	18.6%	0.09
History of myocardial infarction	1.9%	4.7%	0.22
Myocardial_revascularization	4.2%	11.5%	0.03
Heart failure	21.3%	36.9%	0.009
Previous thromboembolic event	6.4%	7%	0.8
Dyslipidemia	53.2%	67.4%	0.07
Type 2 diabetes	11.5%	26.7%	0.0022
Obstructive sleep apnea syndrome	23.6%	37.6%	0.05
Smoking	14.6%	9.6%	0.2
Structural heart disease	7%	22.4%	0.0005
Persistent atrial fibrillation	32.9%	58.6%	<0.0001
Sick sinus syndrome	12.2%	10.1%	0.6
**EKG:**			
LBBB	1.4%	16%	0.0002
RBBB	4.6%	1.4%	0.22
QRS_duration	93.04 ± 16.90	102.06 ± 21.51	0.006
Left atrial enlargement (DII, V1)	29.6%	62.9%	0.001
f-waves > 0.1 mV	71%	27.3%	<0.0001
Left ventricular hypertrophy (LVH)	5.3%	8.3%	0.53
QRS fragmentation	31.3%	32%	0.92
**Echocardiography**			
E/A < 1	24.3%	18.5%	0.53
Indexed LA_volume (mL/m^2^)	33.94 ± 9.33	44.74 ± 14.33	<0.0001
EDLVD (mm)	49.78 ± 4.94	52.77 ± 7.13	0.02
LVEF	56.72 ± 6.67	51.74 ± 10.29	<0.001
Moderate/severe MR	2.7%	21.8%	<0.0001
Moderate/severe aortic stenosis	1%	9.6%	0.01
Moderate/severe TR	4.8%	13.6%	0.001

**Table 2 jcm-13-00061-t002:** Predictors of arrhythmic recurrence according to Cox proportional hazards regression.

Analyzed Variable	B	SE	Wald	Sig.	Exp (B)	95% CI for Exp (B)	
						Lower	Upper
Persistent AF	0.877	0.357	6.021	0.014	2.404	1.193	4.844
f_waves > 0.1 mV	−1.494	0.414	12.992	0.000	0.225	0.100	0.506
LAVI	0.039	0.013	9.214	0.002	1.040	1.014	1.066
Type_2_diabetes	1.038	0.393	6.995	0.008	2.825	1.308	6.098
Height_cm	−0.033	0.018	3.614	0.050	0.967	0.934	1.001

**Table 3 jcm-13-00061-t003:** Prediction power of each variable separately vs. model score (ROC analysis).

	AUC	Std Error	*p* Value	CI	
AF type persistent	0.573	0.031	0.018	0.513	0.634
F waves < 0.1 mV	0.711	0.037	0.000	0.639	0.783
LAVI	0.681	0.037	0.000	0.608	0.754
Type 2 diabetes	0.567	0.032	0.032	0.505	0.630
Height	0.644	0.031	0.000	0.584	0.704
VAT-DHF	0.869	0.034	0.000	0.802	0.936

**Table 4 jcm-13-00061-t004:** The calculus formula for the VAT-DHF score.

LA Volume (indexed)	<28 mL/m^2^	0 points	×1.25	0
	29–33 mL/m^2^	1 point		1.25
	34–39 mL/m^2^	2 points		2.5
	≥40 mL/m^2^	3 points		3.75
AF Type	paroxysmal	0 points	×2	0
	persistent	1 point		2
Type 2 Diabetes	- without type II diabetes			0
	- with type II diabetes			1
Height	- >170 cm			0
	- ≤170 cm			1
f waves	- >1 mm (0.1 mV)		×3	0
	- <1 mm (0.1 mV)			3
Maximum number of points				10.75

## Data Availability

The data presented in this study are available on request from the corresponding author.

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
