# Peer review of "New Score for Predicting Results after Catheter Ablation for Atrial Fibrillation: VAT-DHF"

_jcm, 2023, doi:10.3390/jcm13010061_

Round 1

Reviewer 1 Report

Comments and Suggestions for Authors

1. Please follow the recommendations published by The TRIPOD statement.

2. In line 78, mention the cutoff of the p-value used to include the multivariate regression model, diagnosis of the model, etc.

3. In lines 81-82, Could you explain in more detail how the value is assigned? In the text, it is mentioned "empirically."

4. In Table 2, change Diabetes II to Type 2 Diabetes.

5. In line 125, is the cut-off value 5.8 or 5,875?

6. Remove the table in Figure 3; mention the AUC, p-value, and IC.

7. Improve Figure 3 (not the SPSS version).

8. What about external validation, missing values, etc. Follow the TRIPOD statement (checklist). 

Reviewer 2 Report

Comments and Suggestions for Authors

The authors present the development of a novel algorithm to aid in predicting outcomes of AF ablation.  A few comments:

1.  In the introduction, the authors state that other scores have been developed but their drawback was that they had not been tested in prospective cohorts.  This statement lead me to believe that this study would be a prospective evaluation, which it is not.  

2.  While a scoring system is helpful to predict outcomes, more and more studies suggest the benefits of catheter ablation versus medical therapy.  Do the authors suggest that patients with a higher score should not undergo catheter ablation?

3.  The variables in the scoring system are all known factors increasing the likelihood of recurrence post catheter ablation.  The authors should address how this study adds value over the established studies showing these factors increase recurrence rate.

Comments on the Quality of English Language

Minor editing/proofreading only.  In figures 1 and 2, please remove the legends that appear to be left over from the statistical software.  They are not relevant to the reader.

Round 2

Reviewer 1 Report

Comments and Suggestions for Authors

None

Author Response

Dear Reviewer 1, thank you so much for your suggestions, please see the attachment
